# Clinical Evidence of Circulating Tumor DNA Application in Aggressive Breast Cancer

**DOI:** 10.3390/diagnostics13030470

**Published:** 2023-01-27

**Authors:** Brahim El Hejjioui, Laila Bouguenouch, Moulay Abdelilah Melhouf, Hind El Mouhi, Sanae Bennis

**Affiliations:** 1Biomedical and Translational Research Laboratory, Faculty of Medicine and Pharmacy, Sidi Mohamed Ben Abdellah University, Fez 30050, Morocco; 2Department of Medical Genetics and Oncogenetics, HASSAN II University Hospital, Fez 30050, Morocco; 3Obstetrics Gynecology Department, HASSAN II University Hospital, Fez 30050, Morocco

**Keywords:** circulating tumor DNA (ctDNA), liquid biopsy, breast cancer, cancer diagnosis, tumor heterogeneity, early detection, therapeutic targets and resistance

## Abstract

Breast cancer is clinically and biologically heterogeneous and is classified into different subtypes according to the molecular landscape of the tumor. Triple-negative breast cancer is a subtype associated with higher tumor aggressiveness, poor prognosis, and poor response to treatment. In metastatic breast cancer, approximately 6% to 10% of new breast cancer cases are initially staged IV (de novo metastatic disease). The number of metastatic recurrences is estimated to be 20–30% of all existing breast tumor cases, whereby the need to develop specific genetic markers to improve the prognosis of patients suffering from these deadly forms of breast cancer. As an alternative, liquid biopsy methods can minutely identify the molecular architecture of breast cancer, including aggressive forms, which provides new perspectives for more precise diagnosis and more effective therapeutics. This review aimed to summarize the current clinical evidence for the application of circulating tumor DNA in managing breast cancer by detailing the increased usefulness of this biomarker as a diagnostic, prognostic, monitoring, and surveillance marker for breast cancer.

## 1. Introduction

Breast cancer (BC) is a disease characterized by various clinical behaviors and biological characteristics, making the prediction and management process more challenging [1]. Specific pathological findings and distinct hereditary or somatic genetic alterations appear to be the major factors linked in some way to the risk of developing breast cancer [2].

Breast cancer (BC) is heterogenous, showing variable morphologic and biological features; thus, it has different clinical behaviors and responses to treatment [3]. Based on molecular and histological evidence, BC could be categorized into three groups: (1) BC expressing hormone receptor (estrogen receptor (ER+) or progesterone receptor (PR+)) commonly noted as luminal tumors and are responsive to endocrine therapy, (2) BC expressing human epidermal receptor 2 (HER2+) which is characterized by the overexpression of *HER2* oncogene and is treated with trastuzumab, (3) Triple-negative breast cancer (TNBC) (ER−, PR−, HER2−) subtype, which is associated with high mortality rates and is not responsive to some drug treatment approaches [4,5]. Nowadays, efforts have been focused mainly on a better understanding of triple-negative breast cancer (TNBC) biology since it is the most clinically aggressive group of breast cancers by: (i) the absence of all three immunohistochemical (IHC) biomarkers, (ii) it affects the youngest women, (iii) and has no specific biomarkers. Therefore, the current treatment of TNBC relies largely on chemotherapy and radiotherapy, with no targeted drug yet approved for TNBC [6,7].

In metastatic breast cancer, metastases are heterogeneous with various somatic mutations and molecular alterations. Consequently, it is essential and necessary to identify new biomarkers to guide treatment and improve clinical cancer management [8].

Intratumoral heterogeneity and tumor evolution contribute to treatment failure in cancer patients, which is explained by cellular mutability, cancer cell survival, and drug selection pressures, including adaptive mutability. Thanks to important collaborative efforts of several research entities such as the Cancer Genome Atlas (TCGA), the Human Tumor Atlas Network, the Pan-Cancer Analysis of Whole Genomes Consortium, and the International Cancer Genome Consortium, this intratumoral and intertumoral heterogeneity has been elucidated through single-cell analyses of primary tumors and corresponding metastases, genomic studies of treated and untreated metastatic disease, and investigations of the tumor microenvironment [9].

Therefore, the ideal approach to address the heterogeneity of breast tumors would be to find a method that can capture the entire genetic map of the tumor. Collecting serial biopsies of the tumors and longitudinal monitoring is essential to catch the phylogenetic progress of the tumor. However, it is not feasible to conduct such procedures through tissue biopsy, as they are highly invasive and may be incomprehensible.

Progress and scientific advances in liquid biopsy have raised the hope of detecting temporal and spatial heterogeneity before and after systemic therapy, or when multiple potentially discordant metastases are present [9].

In this review, we will focus on the study of circulating DNA as a liquid biopsy biomarker, highlighting the latest pre-analytical considerations and analytical techniques for the analysis of this biomarker. This review will also focus on the study of the clinical utility and validity of circulating tumor DNA, outlining recent advances and current challenges, particularly in aggressive breast cancer.

## 2. General Overview of Cell-Free DNA

The presence of cell-free DNA (cfDNA) in healthy people’s blood was first reported by Mandel and Metais in 1948 [10], but it was only correlated with cancer in 1977. Leon and Shapiro, using a radioimmunoassay, showed that the serum of cancer patients contained a significantly higher quantity of cell-free DNA [11]. In 1997, Lo’s team demonstrated the presence of circulating fetal DNA in maternal blood during pregnancy [12]; as a result, the first method of diagnosing trisomy 21 was established [13].

Generally, cfDNA consists of short DNA fragments of 160–180 bp, which are the size of a single nucleosome [14]. Based on the most quantitative studies performed so far, cfDNA concentration within healthy people is between 0 and 100 ng/mL of blood [15,16]. The level of cfDNA can be increased with infection [17], stroke [18], trauma, myocardial infarction [19], tissue damage [20], or cancer [11]. In addition to the bloodstream, cfDNA can be extracted from saliva [21], urine [22,23], cerebral spinal fluid [24], and pleural fluid [25].

Although the source of cfDNA is not completely elucidated, various studies report the origin and possible mechanisms of the release of cfDNA [26,27]. Cell-free DNA can be generated from two main sources: dead cells by several mechanisms of cellular degradation such as necrosis and apoptosis or living cells by active release [28]. This source may differ depending on physiological and pathological conditions [27]. For healthy people, cfDNA may originate principally by apoptosis of lymphocytes or other nucleated cells [29,30], this results in high inter-nucleosomal DNA fragmentation. On the other hand, necrosis produces larger cfDNA fragments, higher than 10,000 base pairs (bp) [31]. Many recent studies demonstrate that cfDNA is involved in normal cell function as an intercellular signaling pathway [27], as well as the induction of neutrophil release into the bloodstream to clear bacterial infections [32]. Further studies suggest that cfDNA has other potential biological functions, such as stimulating cell transformation and tumorigenesis in recipient cells [33,34].

The application and potential use of cfDNA as a biomarker achieved the best success in different clinical phases, thus making it a popular and potential target in a wide range of research areas, principally prenatal diagnosis [35,36], organ transplantation monitoring [37,38], and cancer [39,40].

In cancer, every tumor cell potentially releases circulating tumor DNA (ctDNA) [41]. The growing interest in this biomarker is simply due to its potential use as a liquid biopsy. This approach holds great promise in a wide range of clinical applications [42], mainly: early detection of cancer [43], monitoring of tumor dynamics [44], as well as analysis of the evolution of genetic or epigenetic alterations characterizing the tumor [45].

In addition to its non-invasiveness, rapidity, and low cost, ctDNA allows for longitudinal monitoring of cancer in real time and can potentially capture tumor heterogeneity [46,47]. Indeed, ctDNA can be particularly useful when tumor tissue is unavailable or insufficient for testing. Thus, this molecule can be collected at any time and allows for close monitoring [16,48].

In this review, we outlined recent advances and current challenges in the study of circulating tumor DNA, particularly in breast cancer.

## 3. The Importance of Testing for Circulating Tumor DNA

Compared to liquid biopsy, conventional tissue biopsy presents some drawbacks: in addition to its invasiveness [49], it can only provide a static and spatially limited snapshot of the disease at the time of surgery [50,51] which may lead to false-negative results and suboptimal therapeutic selection as a result of their limitation for capturing intratumoral heterogeneity [52]. Given that biopsy samples may be inadequate for routine genetic profiling in up to 30% of cases [53].

Cell-free DNA (cfDNA) analysis, also known as liquid biopsy, created new options for non-invasive diagnosis and therapeutic monitoring, and these liquid biopsies may mirror clinically relevant genetic alterations, which occur in all tumor tissues [41,52,53].

Circulating tumor DNA (ctDNA) represents a substantial fraction of circulating DNA, which varies from <0.05% to 90% depending on tumor location, size, and vascularity, as well as liver and kidney clearance [16,54,55]. Circulating DNA half-life ranges from 16 min to 2.5 h [56] allowing dynamic, real-time monitoring of tumor status as well as rapid evaluation of therapeutic response [57,58]. The origin of ctDNA is not completely elucidated, it may be derived from circulating tumor cells, primary tumors, or metastasis [59,60], which proves that the genomic alteration repertoire of circulating tumor DNA reflects both primary tumors and distant metastatic sites [61]. Several methods have been implemented to identify quantitative and qualitative tumor-specific alterations, such as gene amplification, gene mutations, gene methylation, and microsatellite abnormalities [62,63]. However, these genetic tumor alterations are informative in a variety of applications namely early detection and prevention, minimal residual disease assessment and prognosis, tumor burden monitoring and therapy guidance, as well as relatively non-invasive repeated serial sampling for continuous disease monitoring [64,65].

Studying circulating tumor DNA in breast cancer by determining the heterogeneity of clinically relevant alterations has created new possibilities for non-invasive diagnosis and therapeutic monitoring, ctDNA analysis could potentially reflect the genetic alterations that occur in all breast cancer tissues [52,53].

## 4. Analytical and Clinical Validity of Cfdna

The application and potential use of ctDNA as a biomarker have achieved the best results in different clinical phases, making it a popular and potential target [66]. Therefore, the development of techniques for the detection of genomic variants in ctDNA is increasing in the clinical oncology setting, despite the uncertainties surrounding pre-analytical considerations, analytical validity, clinical validity, and utility.

Pre-analytical considerations are those parameters that influence the quality of cfDNA and are most likely to compromise the success of the analysis. Analytical validity refers to the ability of a test to identify variants of interest accurately and reliably; the test or analysis must be sensitive, specific, and robust. Clinical validity is explained by the ability of the test to detect the presence or absence of a disease state accurately and completely. Regarding clinical utility (final phase), it is reached when there are high levels of evidence confirming that using the test improves patient outcomes compared to not using it [67].

The pre-analytical steps are crucial in the analysis of circulating DNA because it is a very sensitive analysis, since tumor-derived DNA is a small fraction of cfDNA, with possible contamination by leukocyte genomic DNA in the case of leukocyte degradation, which makes a downstream analysis of circulating tumor DNA mutations more difficult and the quantification of this biomarker less accurate.

Plasma is the optimal specimen type for cfDNA analysis, studies showed that the amount of normal DNA derived from leukocyte lysis, which dilutes cfDNA, is much higher in serum than in plasma, this crucial step is excellent to ensure if the plasma is separated from the leukocyte fraction immediately after blood collection or the blood is collected in collection tubes containing a leukocyte stabilizer. The freezing of unfiltered whole blood should not be performed and plasma should be isolated before it is frozen. Although exposure of plasma to a single freeze-thaw cycle does not affect downstream cfDNA analysis, multiple freeze-thaw cycles may lead to nucleic acid degradation. Studies agree that storage of frozen plasma before DNA extraction does not affect further cfDNA analysis.

For the analytical phase, currently, ddPCR is the most widely used method for cfDNA analysis, this technique allows precise detection of known DNA mutations with a better detection limit as well as a more accurate quantification of the ctDNA fraction, the major challenge is the high cost and a low number of installed ddPCR instruments, given that ddPCR is currently capable of analyzing only one potential mutation per reaction [68,69].

However, the limitation of PCR approaches is that these techniques only detect known mutations in certain genes, so patients without these mutations will be overlooked, limiting the application of this technology as a generic diagnostic technique for ctDNA analysis [70].

On the other hand, NGS approaches cover a wider range of mutations by examining the entire gene sequences of interest. However, targeted approaches for ctDNA profiling generally sequence tens of genes to hundreds of genes or even the entire exome, a high sensitivity can then be achieved by deep sequencing of specific regions of interest that cover clinically relevant mutations. Several technologies are struggling to innovate more suitable platforms for ctDNA sequencing, notably sequencing by synthesis (SBS) from Illumina, Ion Torrent from Thermo Fisher Scientific, and nanopore sequencing from Oxford Nanopore Technologies; however, Illumina’s sequencing platform currently dominates due to its high throughput and accuracy [70,71].

While cfDNA NGS sequencing offers a non-invasive approach to the identification of clinically relevant somatic genomic alterations, it is probably not intended to replace the need for tumor biopsies as the gold standard for diagnosis and genotyping, but it can be used as a complementary or alternative analysis when a tissue biopsy is not possible. In addition, ctDNA analysis by NGS is limited by a relative scarcity of total cfDNA and a low fraction of ctDNA (tumor-derived cfDNA fraction), and therefore the development of methods that increase the sensitivity and specificity of cfDNA sequencing is necessary for overcoming these limitations [71].

## 5. Gene Analysis

Several genes present a potential target for studying breast cancer, in this review, we will provide accurate and relevant information on the clinical utility of circulating DNA, we have not targeted all breast cancer genes and signaling pathways, instead we have specifically targeted the most studied genes to evaluate the clinical utility of this ctDNA.

The genes studied have been listed in Table 1, and the signaling pathways of these genes were mentioned in Figure 1.

### 5.1. BRCA1/2 Genes

Breast cancer with *BRCA* mutations is characterized by its aggressiveness, *BRCA1* mutated breast cancer is frequently high grade and triple negative, *BRCA2* related breast cancer is on average of a higher histological grade than sporadic cases [82]. Somatic mutations in *BRCA1/2* occur in approximately 3% of all sporadic breast cancers [83].

The *BRCA1* gene encodes a nuclear phosphoprotein that acts as a tumor suppressor gene by maintaining genomic stability [84]. The encoded protein combines with other tumor suppressors, DNA damage sensors, and signal transducers to form a large multi-unit protein complex known as the *BRCA1* genome-associated surveillance complex [85].

The *BRCA2* gene is involved in the maintenance of genomic stability and more specifically the homologous recombination (HR) pathway that repairs double-stranded DNA breaks. *BRCA2* is located on chromosome 13q12.13, it is a large gene comprising 27 exons coding for 3418 amino acids [86].

In The Cancer Genome Atlas breast cancer study, which performed exome sequencing of tumor and normal samples from a selected cohort of breast cancer patients, a 1/3 somatic to 2/3 germline ratio was found [87] and a similar ratio of somatic to germline mutations has been found in other studies [88].

In a large-scale study in the United States, Vidula et al. analyzed cDNA by NGS and demonstrated that this approach can identify three classes of clinically relevant BRCA1/2 mutations: germline mutations, somatic loss of function, and reversions. This study supports the notion that systematic analysis of cfDNA in patients with advanced cancer can help identify potential candidates for appropriate genotype-based therapy [74].

In line with this, another study conducted in 2017 to analyze cfDNA by massively parallel sequencing, this study targeted all exons of 141 genes and all exons and introns of *BRCA1* and *BRCA2*, as well as functional studies to assess the impact of putative *BRCA1/2* reversion mutations on *BRCA1/2* function, the findings of this study support the utility of cfDNA sequencing to identify putative *BRCA1/2* reversion mutations and facilitate patient selection for PARP inhibition therapy [75].

### 5.2. ESR1 Gene

Over 75% of primary breast cancers are ER-positive, for which hormone therapy is the gold standard in the treatment armamentarium [58,89]. The estrogen receptor (ER) is a transcription factor involved in cell proliferation and activation [90]. Estrogen receptor alpha is encoded by the *ESR1* gene. The acquisition of alterations in this gene has been identified as an important resistance mechanism to endocrine therapy [91]. *ESR1* mutations rarely occur in primary tumors (~1%) but are relatively common (10–50%) in metastatic cancers presenting resistance to endocrine therapy, this is explained by the therapeutic pressure that stimulates the emergence of activating mutations of *ESR1* in the metastatic tissue. These gene mutations are associated with shorter progression-free survival [90]. Therefore, it is important to test for *ESR1* somatic mutations in ctDNA in order to detect the onset of this molecular resistance even before clinical progression. To predict treatment outcomes, because this may dictate patient management, including monitoring and modifications of the treatment plan, eventually this approach can be used to guide sequential treatment options in patients [68,71]. The potential value of screening for *ESR1* mutations in metastatic cancer has increased.

In 2016, a secondary analysis of the *BOLERO-2* clinical trial *(clinicaltrials.gov Identifier: NCT00863655)* was reported in the United States, including 541 patients. *ESR1* gene mutations were analyzed from cfDNA using a droplet digital polymerase chain reaction. The main result of this work was the high prevalence of ER mutations for patients with ER+ metastatic breast cancer treated by aromatase inhibitors (AIs), and the association of this gene’s mutations with more aggressive disease biology, as explained by a median overall survival of 20.73 months (95% CI, 17.71–28.06 months) for patients with ESR1 gene mutations compared with 32.1 months (95% CI, 28.09–36.4 months) for patients without mutations [72].

In a Japanese study in 2017, involving 86 ER+ breast cancer patients, including 69 patients diagnosed with metastatic breast cancer (MBC) and 17 patients with primary breast cancer (PBC), the results showed the absence of *ESR1* mutations in the PBC group, whereas in the MBC group *ESR1* cfDNA mutations were detected in 28.9%. The clinical impact of these mutations was significantly important: all *ESR1* mutation-positive patients had resistance to previous treatments with aromatase inhibitors (AIs) compared to 71.4% of *ESR1* WT patients; 85% of ESR1 mutation-positive patients had resistance to previous treatment with selective estrogen receptor modulators (SERMs) compared to 51% of *ESR1* WT patients. Otherwise, cfDNA monitoring in a subgroup of 52 patients showed that loss of *ESR1* mutations was associated with a longer response time [73].

In January 2020, a Chinese comparative study was conducted comparing 297 tumor samples of primary breast cancer patients and 43 blood samples of metastatic breast cancer (MBC) patients. Next-generation sequencing (NGS) was used by targeting the whole exon of the *ESR1* gene, and the result’s findings are consistent with previous studies: *ESR1* mutations were more frequently detected in circulating tumor DNA of MBC patients than in PBC patients, and the *ESR1* mutation frequency in patients using aromatase inhibitors (AIs) was significantly higher than those who were not using AIs [92].

A French large study targeting twelve genes by whole exome sequencing has identified *ESR1* as a pilot gene specific to MBC, they confirmed that the *ESR1* gene mutation is the most frequent “metastasis-specific” mutation observed in MBC, by identifying that 100% of *ESR1* mutation-positive cases are ER + and present resistance to endocrine therapy [93]. In 2020, a confirmative study by Shibayama and Al reported that *ESR1* mutations were found to be a prognostic predictor of acquired resistance to endocrine therapy [94].

Collectively, these studies demonstrate how the genetic profile of MBC is different from that of primary breast tumors. Hence, metastatic cancer profiling must become a primary step in the definition of optimal treatments for patients; thus, monitoring and surveillance studies will be critical to confirm the effectiveness after treatment. A final point regarding these findings was how easily and feasibly this biomarker could be obtained, plasma samples can be easily collected for ctDNA analysis, allowing for dynamic testing that will increase our understanding of the disease progression and the design of strategies to improve outcomes.

### 5.3. PIK3CA Gene

The *PIK3CA* gene encodes the A isoform of the catalytic subunit (p110a) of the phosphatidylinositol 3-kinases class IA. Phosphatidylinositol 3-kinases (PI3K) are involved in the conversion of phosphatidylinositol 4,5-biphosphate (PIP2) to phosphatidylinositol 3,4,5-triphosphate (PIP3). This PI3K class has a critical role in the control of various cellular processes such as cell growth and proliferation, metabolism, and migration via the PI3K/AKT/mTOR pathway [95].

In approximately 40% of HR + breast cancer cases, the most common molecular alterations are the activating mutations of the PI3K subunit in the *PIK3CA* gene, these mutations induce hyperactivation of the p110α catalytic subunit, leading to constitutive phosphorylation of the AKT and its forward effectors [96,97].

Several preclinical studies indicate that the PIK3/AKT/mTOR pathway alterations may correlate with resistance to CDK4/6 inhibitors (CDK4/6i). Cyclin-dependent kinase 4 and 6 (CDK4/6i) inhibitors have shown clinical efficacy in ER-positive MBC, although their cytostatic effects are limited by primary and acquired resistance, putative mechanisms of resistance to CDK4/6i have been identified [76,98].

An Italian study completed in January 2021 showed that patients with a *PIK3CA* mutation in the blood at the start of CDK4/6i treatment had a significantly shorter Progression Free Survival (PFS) compared to patients without a mutation. Therefore, PI3K status should be considered as a potential predictive biomarker of CDK4/6i resistance. The integration of PI3K status assessment with other molecular information in a surveillance system can improve the accuracy of predicting Overall Survival (OS) and PFS of patients with metastatic breast cancer and may suggest the best treatment strategy. On the other hand, it should be noted that PI3K status as such cannot be the only responsible for CDK4/6i resistance, and thus, special attention should be given to all mutations that also promote the activation of the PI3K/AKT/mTOR signaling pathway [76].

In the same sense, targeted therapy drugs in the final phase of clinical trials, specifically targeting PI3K, are designed to be administered to patients whose tumor has a *PIK3CA* gene alteration, which makes their detection particularly important in tumor genetics.

In July 2017, a Phase I study was designed to evaluate the effectiveness of TASELISIB which is a selective inhibitor of tumor growth by suppression of the PI3K pathway, the results show an increase in the antitumor activity for patients with *PIK3CA* mutant tumors, confirming the results in preclinical trials [99].

In February 2021, the *SANDPIPER* trial *(ClinicalTrials.gov NCT02340221)* (a randomized, multicenter, international, double-blind, placebo-controlled Phase III trial) evaluated the clinical benefit of TASELISIB (a powerful, selective PI3K inhibitor) in combination with FULVESTRANT for Advanced Breast Cancer in a population of 516 *PIK3CA* mutated patients. The results showed a statistically significant improvement in progression-free survival evaluated by the investigator in the *PIK3CA* mutant population [100].

Furthermore, an open-phase 1b study *(ClinicalTrials.gov identifier: NCT01219699)* conducted by Juric et al. aims to evaluate, in patients with advanced ER + breast cancer, the maximum tolerated dose (MTD), safety and activity of L’ALPELISIB which is an oral specific PI3Kα inhibitor, combined with FULVESTRANT, the clinical trial results suggest that the combination of ALPELISIB and FULVESTRANT may have greater clinical activity in *PIK3CA*-altered tumors compared to wild-type tumors [101].

In May 2019, a Phase 3 randomized clinical trial *SOLAR-1 (SOLAR-1 ClinicalTrials.gov number, NCT02437318)* was designed to compare ALPELISIB plus FULVESTRANT with placebo plus FULVESTRANT, this study included 572 patients of which 341 patients had confirmed tumor tissue *PIK3CA* mutations, the results showed a prolongation of progression-free survival in patients with *PIK3CA*-mutated advanced breast cancer who were treated with ALPELISIB-FULVESTRANT [102]. However, these clinical trials highlight the increased utility of testing for *PIK3CA* mutations in circulating tumor DNA as part of daily clinical practice.

### 5.4. TP53 Gene

Tumor suppressor *TP53* is considered the genome gatekeeper. Several studies indicate that *TP53* mutations increase the risk of cancer and, once cancer occurs, these mutations promote invasion, metastasis, and chemoresistance [103]. In invasive breast cancer, *TP53 (p. 53)* is the most frequently mutated gene, which is mutated in approximately 80% of triple-negative breast cancer (TNBC) [104].

The presence or absence of *TP53* mutations is routinely investigated in clinical practice on tumor tissues, using immunohistochemistry or sequencing techniques. However, these methods have various limitations, such as invasiveness and inability to identify tumor heterogeneity and progression, which underline the need for a more robust and sophisticated technology [105]. Currently, *TP53* gene mutations can be detected by cfDNA analysis given the ability of this biomarker to detect tumor heterogeneity, which is a limitation of tumor biopsies [106,107].

In 2017, a French study was realized including 47 patients with non-metastatic triple-negative breast cancer (TNBC) treated with neoadjuvant chemotherapy (NCT), and the study targeted the *TP53* gene at several time points: before NCT; after 1 cycle; before surgery, and after surgery. The results indicated a decrease in ctDNA levels during NCT; however, a small decrease in ctDNA levels during NCT was significantly associated with shorter survival. This research confirmed that the high prevalence of *TP53* mutations in TNBC is a potential biomarker for ctDNA monitoring during NCT, and that ctDNA may become a clinically useful prognostic tool for managing TNBC patients treated with NCT [77].

In July 2019, Savli and colleagues showed that variations in the *TP53* gene are strongly observed in breast cancer samples, this study demonstrated that *TP53* pathogenic variants detection and monitoring by ctDNA analysis is recommended as a useful biomarker for predictive studies, for tumor growth monitoring and personalized treatment strategy planning [78].

Screening for *TP53* mutations in ctDNA can provide monitoring for early detection of genetic events underlying drug resistance and can also inform therapy approaches [79], FEI MA et al., *(ClinicalTrials.gov NCT01937689)* have exploited ctDNA profiling before and after treatment with an oral anti-HER1/HER2 tyrosine kinase inhibitor for 18 HER2-positive metastatic breast cancer patients, and they identified that mutations in *TP53* and PI3K/Akt/mTOR pathway genes were strongly implicated in resistance to HER1/HER2 blocking [108].

The overall clinical trials confirm that *p53* is both a potential prognostic and predictive biomarker and a therapeutic target for breast cancer patients, particularly for TN subtype patients.

### 5.5. ERBB2 Gene

The human epidermal growth factor receptor 2 (*HER2*) is overexpressed in 20–30% of breast cancer patients [109], this amplification is “acquired” in approximately 2–5% of metastatic breast cancers that originally had primary cancers that were not HER2 amplified [110]. Blocking HER2 activity through trastuzumab provides a better outcome for HER2-positive patients. However, resistance in women with metastatic breast cancer (MBC) is usually observed [111].

As targeted anti-HER2 therapies are developed, the assessment of *HER2* status will be more important to stratify patients to the most appropriate treatment regimens [110], and this requires repetitive tumor sampling to identify whether the cancer’s genetic profile has changed after previous treatment [112]. The analysis of circulating DNA has the potential to screen for *HER2 (ERBB2)* amplification acquisition in a non-invasive manner.

In 2017, a phase II study *(Clinicaltrials.gov, NCT01670877)* was conducted to evaluate the clinical benefit rate (CBR) of NERATINIB as monotherapy through determining *HER2* mutations based on circulating DNA analysis, whereby the results indicated the efficacy of NERATINIB in HER2 mutated non-amplified breast cancer and justify that cfDNA sequencing may offer a non-invasive strategy to identify patients with HER2 mutated cancers [80]. In a similar direction, and at a larger scale, a multicenter, open-label, multicohort, phase 2a clinical trial *(ClinicalTrials.gov, NCT03182634)* in 18 UK hospitals was conducted, the results confirm the clinically relevant activity of targeted therapies against rare activating mutations in breast cancer, in HER2 mutant breast cancer *(ERBB2)* identified by cfDNA tests, NERATINIB had comparable activity to that observed when guided by tissue tests [81].

In October 2020, Kleftogiannis et al. developed a new error-correcting cfDNA sequencing approach using bioinformatics strategies to identify tumor-associated genomic alterations. This assay was also performed effectively on the detection of copy number variations (CNVs) in the *ERBB2* oncogene. This result creates opportunities for better tumor characterization, in which sequential plasma samples can be collected to represent CNVs and variants more precisely over time [113].

## 6. Conclusions

Breast cancer displays high levels of heterogeneity and is generally subject to clonal evolution underpinning drug therapy. Through our review, we have demonstrated that the multipoint analysis of cfDNA reflects clonal evolution and allows us to track the molecular landscapes of cancer cell growth by capturing broader molecular alterations that could affect the efficacy of targeted therapies. The shorter execution time of cfDNA analysis and its high sensitivity and specificity are key factors in providing new opportunities for adaptive personalized therapies, optimizing healthcare resources, and enabling higher treatment efficacy and lower risk side effects.

The potential use of cfDNA in the management of breast cancer has been significantly improved by recent advances in molecular technologies, as digital PCR and next-generation sequencing technologies take hold, and as an understanding of the biology and clinical potential of cfDNA increases, the ultimate use of cfDNA in clinical practice seems assured.

## 7. Perspectives

Several clinical trials in Europe, North America, and Asia are underway to evaluate the diagnostic utility of ctDNA, and hence the possibility of incorporating this biomarker into cancer monitoring. Minimally invasive methods of cancer diagnosis have potential because current tumor biopsy and medical imaging techniques that require exposure to ionizing radiation are limited to high-risk individuals and those with previously identified lesions. Alternatively, liquid biopsy-based diagnostics are adapted to repeated sampling and can potentially be used for early detection or screening of cancer.

The challenge is to establish adequate and international pre-analytical conditions, to optimize state-of-the-art analytical techniques with improved sensitivity and specificity, for the purpose of avoiding missing mutations and tumor changes.

## Figures and Tables

**Figure 1 diagnostics-13-00470-f001:**
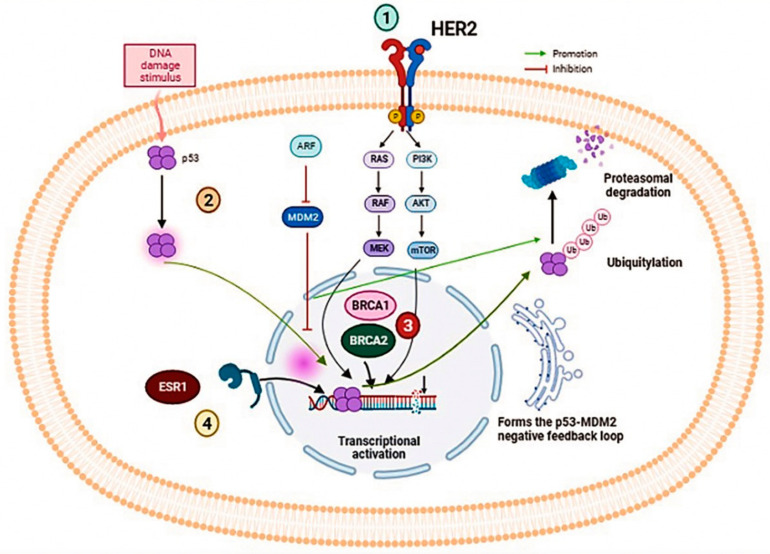
Illustration of several signaling pathways implicated in the oncogenesis of breast cancer. (1) ERBB2 associated protein HER2 signaling pathway, (2) p53 protein implicated in the apoptosis, (3) BRCA1/2 couple implicated in DNA repair mechanisms such as Homologous recombination, (4) ESR1 engaged as a transcription factor.

**Table 1 diagnostics-13-00470-t001:** Summary of molecular and clinical utility of ctDNA in breast cancer.

Gene	Patient Cohort	Molecule Testing Technique	Main Findings	Clinical Significance	References
* **ESR1** *	541 postmenopausal women with a diagnosis of MBC.	Analyzed *ESR1* mutations (Y537S and D538G) on cell-free DNA (cfDNA) using droplet digital polymerase chain reaction (ddPCR).	D538G (21.1%) Y537S (13.3%) 30 had both mutations.	These mutations were associated with shorter overall survival: - wild-type, 32.1 months - D538G, 25.99 months - Y537S, 19.98 months - Both mutations, 15.15 months.	[72]
	86 estrogen receptor-positive BC patients.185 plasma samples (151 plasma samples from 69 MBC patients, and 34 plasma samples from 17 primary BC (PBC) patients).	Multiplex droplet digital PCR assays in a snapshot and serially.	cfDNA *ESR1* and *PIK3CA* mutations were found in 28.9% and 24.6% of MBC patients, respectively.	All patients with *ESR1* mutations had resistance to prior AI (aromatase inhibitor) therapy. 85% of patients with *ESR1* mutations had resistance to prior SERM (Selective estrogen receptor modulators) therapy.	[73]
** *BRCA1/2* **	828 patients with advanced breast, ovarian, prostate, or pancreatic cancer. (the study was conducted in accordance with the Declaration of Helsinki).	Plasma-based NGS assay.	Of 828 patients, 60 (7.2%) had at least one *BRCA1/2* loss-of-function mutation, 42 patients with germline mutations and 18 (14 patients had breast cancer) with somatic mutations only.	NGS analysis of cfDNA identified high rates of therapeutically relevant mutations, including deleterious *BRCA1/2* somatic mutations missed by germline testing.	[74]
	24 patients with proven *BRCA1/2* germline mutations (19 ovarian cancer patients and 5 patients with MBC who received prior treatment with platinum-based chemotherapy and/or PARP inhibitors).	Targeted massively parallel sequencing of tumor DNA from ovarian cancer patients, cfDNA from ovarian and breast cancer patients, and their germline DNA.	Identification of *BRCA1* or *BRCA2* reversion mutations in the cfDNA of 4 ovarian cancer patients (21%) and 2 breast cancer patients (40%).	cfDNA sequencing can help identify putative *BRCA1/2* reversion mutations which may facilitate patient selection for PARP inhibition therapy.	[75]
** *PIK3CA* **	Thirty patients with advanced BC (ABC);	*PIK3CA* mutation analysis was performed using ddPCR.	The presence of a PI3K mutation in liquid biopsy correlates with worse PFS in patients with ABC receiving CDK4/6i.	Integration of PI3K status assessment with other molecular information could improve the management of patients with aggressive breast cancer and better suggest the best therapeutic strategy.	[76]
** *TP53* **	46 patients with nonmetastatic triple-negative breast cancer;	Characterization of TP53 gene mutations in tumor tissue through massively parallel sequencing (MPS). Monitoring of previously characterized mutations based on ctDNA analysis by ddPCR at four time points: pre-NCT, post-cycle, pre-surgery, and post-surgery.	Results show a marked decrease in ctDNA levels and positivity rate during chemotherapy cycles.	The high prevalence of TP53 mutations in TNBC is a potential biomarker for ctDNA monitoring during NCT, and therefore is a tool for TNBC management.	[77]
	113 lung and 18 breast cancer patients	NGS analysis of ctDNA: Panel for hot spot regions in 11 genes for lung cancer and 10 genes for breast cancer.	Variations in the *TP53* gene were detected at a high frequency in both tumor types, followed by the *PIK3CA* gene in breast cancer.	Based on NGS and ddPCR techniques, liquid biopsy could be a very effective method for managing terminal cancer cases and monitoring treatment responses.	[78]
	68 patients with metastatic breast cancer (MBC).	cfDNA and gDNA (Genomic DNA) analysis by next-generation sequencing (NGS)	*TP53* mutations occurred in 10 (45.45%) TNBC patients, 9 (36.00%) HER2+ patients, and 7 (22.22%) HR+ patients. *TP53* represents the gene with the highest number of somatic mutations.	Mutations in *TP53* cDNA and *PIK3CA* genes likely limit survival and promote disease progression.	[79]
** *ERBB2* **	636 women with HER2 nonamplified MBC.	ctDNA analysis by NGS.	Results of this study indicate the efficacy of neratinib for *HER2*-mutated nonamplified breast cancer.	This study supports the potential use of ctDNA to identify patients with *HER2*-mutated breast cancer to establish a new standard of care.	[80]
	Multicohort, phase 2a, platform trial of ctDNA testing in 18 UK hospitals.1051 patients were registered in the study.	ddPCR and NGS are used to detect ctDNA mutations. Patients were recruited into four parallel treatment cohorts corresponding to the mutations identified in the ctDNA (*ESR1; HER2; AKT1* and *PTEN*).	The findings of this study demonstrate the clinically relevant activity of targeted therapies against rare *HER2* and *AKT1* mutations.	The results of this research show that ctDNA analysis, with the technologies used in this study, is accurate enough to be routinely adopted into clinical practice.	[81]

## Data Availability

Not applicable.

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
