# Peer review of "Clinical Evidence of Circulating Tumor DNA Application in Aggressive Breast Cancer"

_diagnostics, 2023, doi:10.3390/diagnostics13030470_

Round 1
Reviewer 1 Report
The authors of this review manuscript are trying to make a case for using ctDNA (a small subset of circulating cell free DNA) to identify changes in tumor mutations to stratify breast cancer patients and allow for personalized treatment strategies. Liquid biopsies would be particularly useful when tissue biopsies are not available and to allow for frequent monitoring time points. The authors make the case that evaluating mutation prevalence in several well known cancer drivers can allow evolving genetic alterations to be used for prognostic and therapeutic determination. There would be no disagreement that mutations in the genes for these proteins (estrogen receptor, PI3K, p53, and HER2) are important drivers of disease progression. A lot of the manuscript is devoted to confirming that mutations are more prevalent in metastatic disease than in the primary tumor, something that most readers at all familiar with breast cancer would know. The authors have reviewed publications and included general, but not specific, statements about how mutation status can be identified in blood samples.What is missing is a summary and clear description of how these studies were performed, challenges encountered and the current state of development of these assays. Missing are references such as this one, "Limitations and opportunities for the analysis of cell-free DNA in cancer diagnostics" Nat Biomed Eng 2022 Mar, 6(3): 232-245. Although this paper may have been published after this manuscript was completed, other papers also exist to remind readers that although there is great interest and promise in the opportunities liquid biopsies present, there are still many challenges on the technical side that are not really addressed here. It must be remembered that the amount of circulating cell-free DNA in blood is small and the amount that was derived from tumor cells is a portion of that, making point mutations difficult to find. There are still challenges in working out sequencing errors that may occur with fragmented DNA. In the conclusion, the authors state what no one would dispute: that breast cancer tumors display high levels of heterogeneity, with clonal evolution as disease progresses. But, what is not stated is how this might be integrated into practice, beyond clinical trials. There is no doubt that there will be continued interest and study that will help, eventually, bring this to the clinic, but no new insights were presented here, beyond providing a not-quite-current literature review of the importance of driver mutations in relevant breast cancer genes.
Author Response
Thank you for the high quality of your reviews, which were very helpful to improve the quality of this review.
We thank you very much for the quality of your reviews, which were very helpful in improving the quality of this article.
Based on your suggestions, we have added a section to discuss pre-analytical considerations, the clinical utility of circulating tumor DNA and current techniques for analysis of this biomarker, as well as the limitations of these techniques and the challenges of developing more sensitive and specific techniques.

Reviewer 2 Report
Reviewer recommendation: major revision
The authors of “Clinical evidence of circulating tumor DNA application in aggressive breast cancer”, made an interesting review of the ctDNA clinical application in breast cancer. As an important component of liquid biopsy, ctDNA analysis could improve current tumor diagnosis and disease monitoring. Moreover, there is no specific biomarker for clinical diagnosis of breast cancer, ctDNA may lead to novel biomarkers for breast cancer in terms of blood based diagnostic technology.
However, some major points need further improvement in this manuscript.
1. The text lacks a description of ctDNA material, isolation, storage, and analysis methods. A scheme summarizing the different material and isolation methods, highlighting the advantage and disadvantages, would be very useful.
2. Figure 1:
Please identify the part of the manuscript that you hope to explain in figure 1. Moreover, how is the y-coordinate axis of Figure 1 plotted? Is there specific data for ctDNA concentration? And if so, please state the reference for the data.
3. Gene analysis: line 114.
The introduction of genes in the article is scattered. It is suggested to make a table to summarize the gene targets, including the literature, target population, gene frequency, reference, and whether it is found in other ctDNA studies of breast cancer. And probably a signature of combination of the associated genes would be more convinced as biomarker application
4. The gene markers described in the review are focused on the frequency, function, and molecular pathway. It is recommended to add the corresponding molecular pathway figures for the specific genes, explaining the distribution of the genetic markers.
5. The conclusion section lacks a perspective on the clinical requirements for gene biomarkers, especially how to get clinical validation and clinical utility for breast cancer patients. The author might comment on this part.
Author Response
Thank you for the high quality of your reviews, which were very helpful to improve the quality of this review.
Based on your suggestions:
Answer 1: We have added a section to discuss pre-analytical considerations and factors that affect the quality and quantity of circulating DNA.
Answer 2:
- We have added a paragraph on the manuscript that explains figure 1.
- Regarding the concentration of circulating DNA, unfortunately, at present there are no studies that indicate precise concentrations of ctDNA specific to each stage of cancer, but there are studies that indicate just the increase and decrease in the ctDNA concentration according to the stage of the disease.
Answer 3: we have added a table that summarizes the genes studied, as well as the research and clinical trials conducted.
Answer 4 : we have added a figure that summarizes the signaling pathways that involve the studied genes.
Answer 5 : We have added a part that talks about the analytical and clinical validity of ctDNA as well as the perspective part that indicates the challenge of developing techniques with higher specificity and sensitivity,

Reviewer 3 Report
This is a peer-review on “Clinical evidence of circulating tumor DNA application in aggressive breast cancer” by Hejjioui and colleagues. The authors provide a short review on liquid biopsy and how to apply this in breast cancer diagnostics and clinical management of breast cancer patients. The manuscript includes major references to this rapidly evolving field and also mentions ongoing recent clinical studies. For a short informative review it is fully acceptable to restrict the references, instead of overloading the reader with too many other studies.
The peer-reviewer is not a native English speaker, but suggests the paper is acceptable with only minor changes, mostly language and punctuation editing; some very minor remarks and suggestions are included in the corrected PDF file.

Author Response
Thank you for all your suggestions and relevant corrections.
Based on your comments, we have tried to correct the modifications that you have highlighted in the whole manuscript.

Reviewer 4 Report
I would suggest the authors to do a major revision and particularly to improve the manuscript quality in the following points:
1. Authors failed to introduce current challenges in measuring ctDNA and new technical advances
2. Authors should clarify the BC stage (primary vs, metastatic) and types (HER+, ER+, triple negative) more in detail when mentioning clinical importance of cfDNA.
3. In part 3, authors focus on some specific genes, however authors failed to cover tumour mutation load, other gene alterations
4. Authors should also cover the clinical importance of cfDNA concentration (prognostic value and predictive value)
Author Response
Thank you for the quality of your reviews, which have been very useful in improving the quality of this review.
Based on your reviews and suggestions, we have made the following changes:
Response 1: we have added a section about the current challenges of cDNA measurement, discussing pre-analytical considerations, the analytical techniques used and their advantages and disadvantages.
Response 2: In the introduction, we have added a section to detail the subtypes of breast cancer, as well as the aggressiveness of each subtype, and in the gene analysis part, we have indicated the specific subtypes targeted.
response 3 : we have not targeted all breast cancer genes and signaling pathways, instead we have specifically targeted the most studied genes to evaluate the clinical utility of ctDNA, since there are other genes related to breast cancer, but there is a lack of ctDNA studies of these genes.
According to your suggestion, we have attempted to search for more genes that we have added to the manuscript.
Response 4: Based on your suggestions, we have discussed the clinical importance of measuring cfDNA concentration, as well as the limitations of measurements. In figure 1, we have also tried to discuss the importance of measuring cfDNA concentration in order to monitor the disease development.

Round 2
Reviewer 1 Report
Although this is a revised manuscript, many of the "revised" sections seem to be identical to the original text except for changes in citation number. The manuscript needs to be edited for sentence structure, typos, syntax. In particular, Sections 4 and 5.1 need editorial attention. For example, the title of section 4 may have a typo. Sometimes ctDNA, referring to circulating tumor DNA, is written as tcDNA. NGS is written as New-generation rather than Next-generation sequencing in one instance.
Gene names need to be in italics, but proteins should be written with no italics. This should be checked in all of the sections.
The authors improved the manuscript by adding a Perspectives section, but this review, while extensively summarizing publications describing the use of cfDNA, do not include information on how these assays are validated....for example, how well are mutations detected? Are test samples "spiked" with target sequences and the limits of detection determined and compared to what would be found in patient samples? Looking for the amount of cfDNA as a marker for tumor response to treatment relies on a much larger amount of target than trying to find a small amount of a specific gene target fragment. In this regard, Section 4 is somewhat lacking in detail.
The authors also include a diagram of HER2 signalling pathways, but this is not really relevant to the review of the feasibility of using plasma rather than biopsy tissue for diagnostics, given that HER2 is already accepted as an important gene in breast cancer classification.
Figure 1. The purpose of this figure seems to be to show that cfDNA is low after initial treatment but may increase after relapse. It is not explained how metastases, which are usually small at relapse, are responsible for more circulating cfDNA than the original tumor, likely of larger size than mets. Also, after successful treatment involving neo or adjuvant therapy, the death of residual cells post surgery would likely increase cfDNA and it has been proposed that this would be a marker for treatment response. Also, since the source of the cfDNA detected at relapse is not known (from local, regional, or distant sites), how the information is used to develop treatment plans. Acquired mutations in cfDNA from the mets may be present in lower amounts in the plasma than if there is cfDNA released into the blood from locally recurrent disease that has not acquired mutations.
Section 5.1 describes the use of BRCA1/2 (the title of this section inexplicably is written as BRCA1 et BRCA2) as a diagnostic tool. It is a bit confusing when they discuss the Vidula (name should be capitalized) paper and how germline mutations are missed, but high levels of ctDNA are found. This editorial in the Society of Clinical Oncology journal helps put the concept into clinical perspective: Genesan, 2018, JCO Precision Oncology, Tumor Suppressor Tolerance: Reversion Mutations in BRCA1 and BRCA2 and Resistance to PARP Inhibitors and Platinum.
The table with the list of clinical trials sorted by gene target would be more helpful if citation numbers were included.
Author Response
I am pleased to resubmit, on behalf of the co-authors, the revised version of the paper entitled “Clinical evidence of circulating tumor DNA application in aggressive breast cancer”.
We are thankful for the constructive comments that helped to considerably improve the quality of the manuscript. We have carefully revised the present paper in the light of your suggestions and replied to each of your concerns.
we have rectified the language and writing errors.
Concerning the table 1: we have added all the parts listed below:
- PATIENT COHORT
- MOLECULE TESTING TECHNIQUE
- MAIN FINDINGS
- CLINICAL SIGNIFICANCE
in part 4: we have given the pre-analytical and analytical considerations. But in table 1 we have tried to give more details about the techniques used and the validity of these techniques according to the researchers of the mentioned studies and clinical trials.
Regarding figure 1, we have removed this figure because we have not found articles that mainly study the concentrations of cfDNA according to the stages of the disease by giving numbers. Instead, we have found studies that compare the concentrations between healthy people, people with a localised cancer, as well as people with a metadtatic cancer.
regarding section 5.1: we have tried to make the requested modifications.
Finally, we would like to thank the Editor-in-chief and Reviewers for considering this revised version, with the hope that the rectifications and answers meet your expectations.

Reviewer 4 Report
Dear Author,
Despite the revised version, the manuscript still has some language and writing errors. Importantly, Table 1 include some studies on those BC "VIP" genes( disappointing only study and trial name), while did not include real findings and clinical significances. I would suggest author to reorganise Table 1 to include main findings and patient cohort, molecule testing technique and other informations. Also at same time author can shorten the text on each gene to keep concise.
Author Response
I am pleased to resubmit, on behalf of the co-authors, the revised version of the paper entitled “Clinical evidence of circulating tumor DNA application in aggressive breast cancer ”.
We are thankful for the constructive comments that helped to considerably improve the quality of the manuscript. We have carefully revised the present paper in the light of your suggestions and replied to each of your concerns.
we have rectified the language and writing errors.
Concerning the table 1: we have added all the parts listed below:
- PATIENT COHORT
- MOLECULE TESTING TECHNIQUE
- MAIN FINDINGS
- CLINICAL SIGNIFICANCE
Finally, we would like to thank the Editor-in-chief and Reviewers for considering this revised version, with the hope that the rectifications and answers meet your expectations.
